# Estimating Propensity for Causality-based Recommendation without Exposure Data

**Zhongzhou Liu**
School of Computing and Information Systems
Singapore Management University
Singapore, 178902
zzliu.2020@phdcs.smu.edu.sg

**Yuan Fang**[*]
School of Computing and Information Systems
Singapore Management University
Singapore, 178902
yfang@smu.edu.sg

**Min Wu**
Institute for Infocomm Research
A*STAR
Singapore, 138632
wumin@i2r.a-star.edu.sg

## Abstract

Causality-based recommendation systems focus on the causal effects of user-item interactions resulting from item exposure (i.e., which items are recommended or exposed to the user), as opposed to conventional correlation-based recommendation. They are gaining popularity due to their multi-sided benefits to users, sellers and platforms alike. However, existing causality-based recommendation methods require additional input in the form of exposure data and/or propensity scores (i.e., the probability of exposure) for training. Such data, crucial for modeling causality in recommendation, are often not available in real-world situations due to technical or privacy constraints. In this paper, we bridge the gap by proposing a new framework, called Propensity Estimation for Causality-based Recommendation (PROPCARE). It can estimate the propensity and exposure from a more practical setup, where only interaction data are available *without* any ground truth on exposure or propensity in training and inference. We demonstrate that, by relating the pairwise characteristics between propensity and item popularity, PROPCARE enables competitive causality-based recommendation given only the conventional interaction data. We further present a theoretical analysis on the bias of the causal effect under our model estimation. Finally, we empirically evaluate PROPCARE through both quantitative and qualitative experiments.

## 1 Introduction

Recommendation systems have been widely deployed in many real-world applications, such as streaming services [34, 5], online shopping [17] and job searching [19]. The primary aim of recommendation systems, such as boosting sales and user engagement [10], depends heavily on user interactions, such as clicking on or purchasing items. Hence, a classical paradigm is to predict user-item interactions, and accordingly, recommend items with the highest probability of being interacted (e.g., clicked or purchased) to users [8, 22, 33, 35]. This paradigm ignores the causal impact behind recommendation [31]: If an item already has a high probability of being interacted by a user without being recommended, *is there really a need to recommend the item to this user?*

---

[*]Corresponding author

37th Conference on Neural Information Processing Systems (NeurIPS 2023).

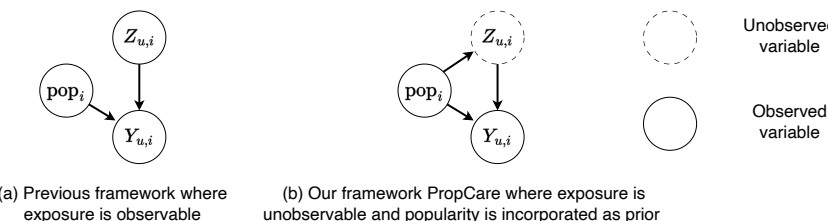

Figure 1: Causal diagrams under different frameworks. $\mathrm{pop}_i$ is the popularity (prior) of item $i$. $Y_{u,i}$ indicates if user $u$ interacts with item $i$. $Z_{u,i}$ indicates if item $i$ is exposed to user $u$.

Recently, a few studies [38, 30, 28, 37] have shifted the focus to this question. They aim to recommend an item based on the uplift, also called the *causal effect*, in the user's behavior (e.g., clicks or purchases) caused by different treatments (i.e., recommending/exposing the item or not) [9]. Such causality-based recommendation systems posit that recommending items with a higher causal effect carries greater merit than those with a higher interaction probability. Typical approaches involve quantifying the causal effect in the user's behavior, based on the observed data and the counterfactual treatment [37]. Existing works assume that the exposure data (i.e., whether an item has been recommended to a user or not), or the propensity scores [25] (i.e., the probability of recommending/exposing an item to a user), are observable at least during the training stage. However, in real-world scenarios, those data are often unavailable. For instance, while it is feasible to log each user who purchased an item in a e-commerce platform, it may be difficult to distinguish between purchases made with or without exposure due to technical and privacy constraints in determining if a user has been exposed to the item a priori. Without the exposure data and/or propensity scores provided during training, existing causality-based recommenders cannot be deployed.

Toward practical causality-based recommendation, we consider a more relaxed and realistic setup where exposure and propensity scores are not observable. Although some previous works [38, 42, 1, 14, 21] have attempted to estimate propensity scores in a different context (e.g., addressing biases in recommendation), they further suffer two key limitations. First, most state-of-the-art methods still require exposure data to train the propensity estimator [38, 1, 14]. Second, they fail to integrate prior knowledge into the propensity estimator, resulting in less robust estimation. To address these challenges and bridge the data gap in many recommendation scenarios and benchmarks, we propose a novel framework of **Prop**ensity Estimation for **Ca**usality-based **Re**commendation (PROPCARE), to estimate the propensity score and exposure of each item for each user. Specifically, we observe a pairwise characteristic that relates propensity scores and item popularity when the probability of user-item interaction is well controlled. (The observation is formalized as Assumption 1 and empirically validated in Sect. 4.2.) Based on the observation, we incorporate item popularity as prior knowledge to guide our propensity estimation. Furthermore, we present a theoretical analysis on the bias of the estimated causal effect. The analysis enables us to investigate the factors that influence our estimation and subsequently guide our model and experiment design.

In summary, we compare previous propensity estimation and PROPCARE in Fig. 1, highlighting our key advantages: PROPCARE does not need propensity or exposure data at all, and incorporates prior information for robust estimation. The contributions of this paper include the following. (1) Our proposed framework bridges the gap in existing causality-based recommendation systems, where the propensity score and/or exposure data are often unavailable but required for model training or inference. (2) We incorporate the pairwise relationship between propensity and item popularity as prior knowledge for more robust propensity estimation. We present a further analysis on the factors that influence our model. (3) We conduct extensive experiments to validate the effectiveness of PROPCARE through both quantitative and qualitative results.

## 2 Related Work

**Causal effect estimation in recommendation**  While typical recommendation systems consider positive feedback or interactions like clicks and purchases as successful, it may be more beneficial to optimize the uplift in interactions, also called the causal effect, solely caused by recommendations [18]. However, obtaining the causal effect in real-world scenarios is challenging because of its

counterfactual nature [9]. Conducting online A/B tests to compare exposure strategies may be feasible but expensive and susceptible to selection bias [27]. To address these issues, several causal effect estimators have been proposed. The naïve estimator [30] assumes random exposure assignment to all user-item pairs, which is inconsistent with most recommendation scenarios. The inverse propensity score (IPS) estimator [30] incorporates the propensity score, defined as the probability of exposure [25], to overcome this limitation. Direct model estimators like CausCF [38] directly predict outcomes using parametric models based on different exposure statuses. A recently proposed doubly robust estimator [37] integrates a parametric model with the non-parametric IPS estimator for reduced bias and variance. However, these estimators require access to input data containing propensity scores and/or exposure data, at least during the training stage, which are often unavailable due to technical and privacy limitations.

**Propensity estimation in recommendation**   Existing causal effect estimation approaches require exposure data and/or propensity scores at least in training, which are frequently unavailable or subject to the missing-not-at-random (MNAR) issue [32]. Hence, we have to rely on their estimations. Some methods estimate propensity in a heuristic way, such as using item popularity [30] or other side information (e.g., items participating in promotional campaigns) [28]. However, these estimations lack personalization and may result in noisy results. Other approaches utilize interaction models (also called click models) [24, 3, 42, 21] to relate propensity scores, relevance and interactions. However, without additional constraints, the interaction model alone can be difficult to optimize as we will elaborate in Sect. 4.1. Besides, matrix factorization [16, 38], linear regression [28], dual learning [21] and doubly robust learning [14] can also learn propensity scores, but they assume exposure data as training labels or known variables, which is incompatible with our setup without any observable propensity or exposure data.

## 3   Preliminaries

**Data notations**   Consider a typical recommendation dataset that contains only interactions between users and items, such as purchases or clicks. Let $Y_{u,i} \in \{0, 1\}$ denote the observed interaction between user $u \in \{1, 2, \ldots, U\}$ and item $i \in \{1, 2, \ldots, I\}$. $D = \{(Y_{u,i})\}$ denotes the collection of observed training user-item interaction data. Note that our framework does not assume the availability of any additional data except the interaction data. Moreover, let $Z_{u,i} \in \{0, 1\}$ denote an *unobservable* indicator variable for exposure, i.e., $Z_{u,i} = 1$ iff item $i$ is exposed/recommended to user $u$. We use $p_{u,i}$ to represent the propensity score, which is defined as the probability of exposure, i.e., $p_{u,i} = P(Z_{u,i} = 1)$.

**Causal effect modelling**   Let $Y_{u,i}^0$ and $Y_{u,i}^1 \in \{0, 1\}$ be the potential outcomes for different exposure statuses. Specifically, $Y_{u,i}^1$ is defined as the interaction between user $u$ and item $i$ when $i$ has been exposed to $u$. Accordingly, $Y_{u,i}^0$ is the interaction when $i$ has not been exposed to $u$. This setup assumes a counterfactual model: In the real world only one of the scenarios can happen, but not both. Subsequently, the causal effect $\tau_{u,i} \in \{-1, 0, 1\}$ is defined as the difference between the two potential outcomes [26], i.e., $\tau_{u,i} = Y_{u,i}^1 - Y_{u,i}^0$. In other words, $\tau_{u,i} = 1$ means recommending item $i$ to user $u$ will increase the interaction between $u$ and $i$ and $\tau_{u,i} = -1$ means the opposite. $\tau_{u,i} = 0$ means recommending or not will not change the user's interaction behavior. Naturally, users, sellers and platforms could all benefit from recommendations that result in positive causal effects.

**Causal effect estimation**   The causal effect cannot be directly computed based on observed data due to its counterfactual nature. Among the various estimators introduced in Sect. 2, direct parametric models [38, 37] are sensitive to the prediction error of potential outcomes [37]. Hence, high-quality labeled exposure data are required in parametric models, which is not the setup of this work. To avoid this issue, we adopt a nonparametric approach, known as the inverse propensity score (IPS) estimator [30], for causal effect estimation as follows.

$$\hat{\tau}_{u,i} = \frac{Z_{u,i} Y_{u,i}}{p_{u,i}} - \frac{(1 - Z_{u,i}) Y_{u,i}}{1 - p_{u,i}}. \tag{1}$$

**Interaction model** In line with prior works [21, 42, 39], we adopt an interaction model [1] [24, 3] that assumes the following relationship between interactions, propensity and relevance:

$$y_{u,i} = p_{u,i} r_{u,i}, \qquad (2)$$

where $y_{u,i} = P(Y_{u,i} = 1)$ is the probability of interaction between user $u$ and item $i$, and $r_{u,i}$ represents the probability that item $i$ is relevant to user $u$.

# 4 Proposed Approach: PROPCARE

In this section, we introduce our propensity estimation approach PROPCARE. We start with a naïve approach, followed by our observation on prior knowledge, before presenting the overall loss for propensity learning and how the learned propensity can be used for causality-based recommendation. We end the section by discussing a theoretical property of our estimation.

## 4.1 Naïve propensity estimator

The overall objective is to estimate propensity scores and exposure from a more practical setup where only interaction data are observable. Since the propensity score $p_{u,i}$ is the probability of exposure $P(Z_{u,i} = 1)$, we focus on the estimation of propensity scores, whereas the corresponding exposure can be readily sampled based on the propensity. The interaction model in Eq. (2) intuitively leads us to the naïve loss function below.

$$\mathcal{L}_{\text{naïve}} = -Y_{u,i} \log f_p(\mathbf{x}_{u,i}; \Theta_p) f_r(\mathbf{x}_{u,i}; \Theta_r) - (1 - Y_{u,i}) \log(1 - f_p(\mathbf{x}_{u,i}; \Theta_p) f_r(\mathbf{x}_{u,i}; \Theta_r)), \quad (3)$$

where $\mathbf{x}_{u,i} = f_e(u, i; \Theta_e)$ is a joint user-item embedding output by a learnable embedding function $f_e$; $f_p$ and $f_r$ are learnable propensity and relevance functions to produce the estimated propensity score $\hat{p}_{u,i}$ and relevance probability $\hat{r}_{u,i}$, respectively. Note that each learnable function $f_*$ is parameterized by $\Theta_*$, and we implement each as a multi-layer perceptron (MLP).

However, through the naïve loss we cannot learn meaningful propensity and relevance functions ($f_p$ and $f_r$), since they are always coupled in a product and can be collapsed into one function. It is equivalent to learning a single interaction function, instead of learning each individual factor.

## 4.2 Incorporating prior knowledge

To avoid the above issue, one solution is to introduce prior knowledge to further constrain the propensity or relevance function. In particular, it has been observed that *more popular items will have a higher chance to be exposed* [41]. The popularity of item $i$, $\text{pop}_i$, is defined based on the total number of observed interactions in the dataset, i.e., $\text{pop}_i = \sum_{u=1}^{U} Y_{u,i} / \sum_{j=1}^{I} \sum_{u=1}^{U} Y_{u,j}$. However, this observation [41], while intuitive, is not adequate in explaining the relationship between popularity and exposure. In particular, items with a higher interaction probability also tend to have a higher chance to be exposed, especially when prior exposure was decided by recommenders in the classical paradigm. To incorporate popularity as a prior toward propensity/exposure estimation, we propose to introduce a control on the interaction probability, and formulate the following assumption.

**Assumption 1 (Pairwise Relationship on Popularity and Propensity)** *Consider a user $u$ and a pair of items $(i, j)$. Suppose the popularity of item $i$ is greater than that of $j$, and their interaction probabilities with user $u$ are similar. Then it follows that item $i$ is more likely to be exposed to user $u$ than item $j$ is.* □

The intuition is that, when a user's interaction probabilities are similar toward two items $i$ and $j$, but item $i$ is more likely to be exposed to the user, the reason could be item $i$ is more popular than $j$. Our assumption essentially places a control on the interaction probability to eliminate its influence on the exposure, and simultaneously isolate the effect of popularity on the exposure.

**Empirical validation of Assumption 1** In the following, we examine our assumption by calculating the fraction of item pairs that satisfy this assumption in three datasets, namely, DH_original,

---

[1]Also called the "click" model when the interaction refers to click in some literature.

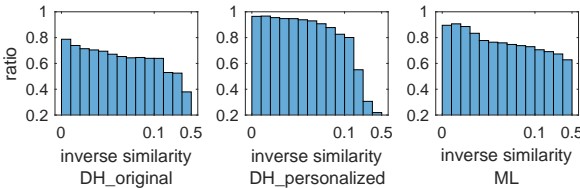

Figure 2: Histogram of item pairs $(i, j)$ that satisfy Assumption 1. The bins are based on the inverse similarity in interaction probabilities, $|y_{u,j} - y_{u,i}|$, divided by $\{0, 0.01, \ldots, 0.09, 0.1, 0.2, \ldots, 0.5\}$. That is, the first 10 bins have an equal width of $0.01$ and the last 4 bins have an equal width of $0.1$.

DH_personalized and ML (see Sect. 5.1 for dataset descriptions). Specifically, we first estimate the probability $y_{u,i}$ of each interaction $Y_{u,i}$ using logistic matrix factorization [11]. We also obtain the propensity score $p_{u,i}$ from ground truth values provided by the datasets (note that we only use the ground truth for evaluation purposes, not in model training or inference). Then, for each user $u$, we place an item pair $(i, j)$, where a randomly sampled $i$ is paired with each of the remaining items, into several bins based on $i$ and $j$'s similarity in their interaction probability with $u$. More specifically, each bin $b$ contains $(i, j)$ pairs such that $|y_{u,j} - y_{u,i}|$ falls into $b$'s boundaries. Finally, we compute the ratio of $(i, j)$ pairs consistent with Assumption 1 to the total pairs in each bin $b$, as follows.

$$\text{ratio}_b = \frac{1}{U} \sum_{u=1}^{U} \frac{\text{\# item pairs } (i, j) \text{ for user } u \text{ in bin } b \text{ s.t. } (p_{u,j} - p_{u,i})(\text{pop}_j - \text{pop}_i) > 0}{\text{\# item pairs } (i, j) \text{ sampled for user } u \text{ in bin } b}. \quad (4)$$

We report the ratios in Fig. 2. It can be observed that when $|y_{u,j} - y_{u,i}|$ is smaller (i.e., $i$ and $j$'s interaction probabilities with $u$ are more similar), a higher fraction of items pairs in the bin satisfy our assumption. In contrast, when $|y_{u,j} - y_{u,i}|$ grows larger (i.e., the interaction probabilities are not well controlled and become less similar), the validity of the original observation [41] becomes weaker. In summary, the results on the three datasets demonstrate the validity of Assumption 1.

**Integrating prior knowledge** Based on Assumption 1, we utilize item popularity to inject prior knowledge on the probability of exposure (i.e., propensity score) through the following loss.

$$-\log\left[\sigma(f_p(\mathbf{x}_{u,i}) - f_p(\mathbf{x}_{u,j}))\right] \text{ s.t. } \text{pop}_i > \text{pop}_j, \; y_{u,i} \approx y_{u,j}, \quad (5)$$

where $\sigma$ is the sigmoid activation and $y_{u,i}$ is computed as $f_p(\mathbf{x}_{u,i})f_r(\mathbf{x}_{u,i})$. While Eq. (3) models propensity in a point-wise manner, Eq. (5) incorporates popularity as prior knowledge in a pairwise manner. The advantage is twofold. First, it decouples the propensity and relevance functions, using only item popularity which can be readily computed from the interaction data shown earlier without the need for external information. Second, by separating the estimated propensity of less popular items and more popular items, it prevents all predicted values from clustering in a narrow range near 1 or 0. This is beneficial in mitigating the issue of high variance caused by extreme values [37].

To materialize the control $y_{u,i} \approx y_{u,j}$ on the interaction probabilities in Eq. (5), we adopt the following loss that involves a soft version of $y_{u,i} \approx y_{u,j}$.

$$\mathcal{L}_{\text{pop}} = -\kappa_{u,i,j} \log\left[\sigma(\text{sgn}_{i,j} \cdot (f_p(\mathbf{x}_{u,i}) - f_p(\mathbf{x}_{u,j}))) + \sigma(\text{sgn}_{i,j} \cdot (f_r(\mathbf{x}_{u,j}) - f_r(\mathbf{x}_{u,i})))\right], \quad (6)$$

where $\text{sgn}_{i,j} \in \{1, -1\}$ is the sign of $(\text{pop}_i - \text{pop}_j)$ and $\kappa_{u,i,j}$ is a weighting function such that it will assign a higher weight if $y_{u,i}$ and $y_{u,j}$ are closer. Specifically, we choose $\kappa_{u,i,j} = e^{\eta(y_{u,i} - y_{u,j})^2}$, where $\eta < 0$ is a learnable parameter. Moreover, according to the interaction model in Eq. (2), for a fixed $y_{u,i}$, a higher $p_{u,i}$ implies a lower $r_{u,i}$. This explains the additional constraint on the relevance function $f_r$ in Eq. (6), which will further improve model training.

### 4.3 Propensity learning

Based on the discussions in Sect. 4.1–4.2, the naïve loss essentially optimizes the interaction model, whereas the pairwise loss utilizes popularity as prior information for propensity learning. For more robust learning, we further take a global view on the distribution of propensity scores, which usually follow a long-tailed distribution [40, 42]. In particular, we employ a beta distribution to regularize the

propensity scores, as has been done in literature in modeling propensity or other long-tailed quantities [4, 15]. Overall, we minimize the following loss toward propensity learning:

$$\min_{\Theta} \mathcal{L} = \sum_{u,i,j} (\mathcal{L}_{\text{naïve}} + \lambda \mathcal{L}_{\text{pop}}) + \mu \text{KL}(Q \| \text{Beta}(\alpha, \beta)). \quad (7)$$

Here $Q$ is the empirical distribution of all estimated propensity scores $\hat{p}_{u,i}$. $\text{Beta}(\alpha, \beta)$ is a reference beta distribution with parameters $\alpha$ and $\beta$ which are selected to simulate a long-tailed shape. $\text{KL}(\cdot \| \cdot)$ computes the Kullback-Leibler divergence between two distributions. $\lambda$ and $\mu$ are trade-off hyperparameters to balance different terms.

Finally, we use the estimated propensity score $\hat{p}_{u,i}$ to predict the exposure variable $Z_{u,i}$: $\hat{Z}_{u,i} = 1$ if $\text{Norm}(\hat{p}_{u,i}) \geq \epsilon$, and 0 otherwise, where $\epsilon$ is a threshold hyperparameter and $\text{Norm}$ is a normalization function such as $Z$-score normalization. The overall training steps are sketched in Algorithm 1 in Appendix A.

### 4.4 Causality-based recommendation

We resort to DLCE [30], a state-of-the-art causality-based recommender equipped with an IPS estimator. It takes interaction $Y_{u,i}$, exposure $Z_{u,i}$ and propensity $p_{u,i}$ as input, and outputs a ranking score $\hat{s}_{u,i}$ for each user-item pair. Given a triplet $(u, i, j)$ such that $u$ is a user and $i \neq j$ are randomly sampled from the item set, the loss of DLCE is defined as follows [30].

$$\frac{Z_{u,i} Y_{u,i}}{\max(p_{u,i}, \chi^1)} \log \left(1 + e^{-\omega(\hat{s}_{u,i} - \hat{s}_{u,j})}\right) + \frac{(1 - Z_{u,i}) Y_{u,i}}{\max(1 - p_{u,i}, \chi^0)} \log \left(1 + e^{\omega(\hat{s}_{u,i} - \hat{s}_{u,j})}\right), \quad (8)$$

where $\chi^1, \chi^0$ and $\omega$ are hyperparameters. We follow the standard training procedure of DLCE, except that we substitute the ground-truth exposure and propensity score with our estimated values $\hat{Z}_{u,i}$ and $\hat{p}_{u,i}$, respectively, in the above loss. Hence, the entire training process for our propensity learning and DLCE do not require any ground-truth exposure or propensity data. After DLCE is trained, for each user $u$, we generate a ranked list of all items based on the optimized $\hat{s}_{u,i}$.

### 4.5 Theoretical property

The performance of causality-based recommendation depends on how accurate we can model the causal effect in the user-item interactions. Although it has been established elsewhere [30] that the IPS estimator defined in Eq. (1) is unbiased as long as exposure $Z_{u,i}$ and propensity score $p_{u,i}$ are correctly assigned, in our setup only estimated propensity scores and exposure are available. Thus, we characterize the bias of the IPS estimator when estimations are used instead.

**Proposition 1** *Suppose we replace the ground truth values of $Z_{u,i}$ and $p_{u,i}$ with the estimated $\hat{Z}_{u,i}$ and $\hat{p}_{u,i}$ in Eq. (1), respectively. Then, the bias of the estimated causal effect $\hat{\tau}_{u,i}$ is*

$$\left( \frac{p_{u,i} + \mathbb{E}\left[\hat{Z}_{u,i} - Z_{u,i}\right]}{\hat{p}_{u,i}} - 1 \right) Y_{u,i}^1 - \left( \frac{1 - p_{u,i} - \mathbb{E}\left[\hat{Z}_{u,i} - Z_{u,i}\right]}{1 - \hat{p}_{u,i}} - 1 \right) Y_{u,i}^0. \quad (9)$$

$\square$

We defer the proof to Appendix B. From the bias stated in Proposition 1, we make two further remarks to guide the learning and evaluation of propensity scores and exposure.

**Remark 1** *The bias is influenced by three major factors: $p_{u,i}/\hat{p}_{u,i}$, $(1 - p_{u,i})/(1 - \hat{p}_{u,i})$ and $\mathbb{E}\left[\hat{Z}_{u,i} - Z_{u,i}\right]$. Note that if $\hat{p}_{u,i} = p_{u,i}$ and $\hat{Z}_{u,i} = Z_{u,i}$, the bias would be zero which is consistent with earlier findings [30].* $\square$

**Remark 2** *If the estimated $\hat{p}_{u,i}$ is extremely close to 0 or 1, the bias can be potentially very large.* $\square$

The above proposition and remarks shed some light on what we should focus on when estimating or evaluate exposure and propensity score. On the one hand, since exposure is a binary variable and the bias is influenced by $\mathbb{E}\left[\hat{Z}_{u,i} - Z_{u,i}\right]$, we may evaluate it with binary classification metrics such as F1 score. On the other hand, since propensity is a continuous variable and estimations extremely close to zero or one should be avoided, regularizing the global distribution in Eq. (7) and ensuring a proper scale of the propensity scores would be useful.

Table 1: Statistics of datasets.

| Dataset | #users | #items | $\bar{Y}_{u,i}$ | $\bar{Z}_{u,i}$ | $\bar{\tau}_{u,i}$ | $\bar{p}_{u,i}$ |
|---|---|---|---|---|---|---|
| DH_original | 2,309 | 1,372 | .0438 | .6064 | .0175 | .2894 |
| DH_personalized | 2,309 | 1,372 | .0503 | .6265 | .0178 | .4589 |
| ML | 943 | 1,682 | .0676 | .0593 | .0733 | .0594 |

# 5 Experiment

In this section, we comprehensively evaluate the effectiveness of the proposed PROPCARE through both quantitative and qualitative experiments.

## 5.1 Experiment setup

**Datasets**    We employ three standard causality-based recommendation benchmarks. Among them, **DH_original** and **DH_personalized** are two versions of the DunnHumby dataset [30], which includes purchase and promotion logs at a physical retailer over a 93-week period. The difference in the two versions mainly lies in the derivation of ground-truth propensity scores as stated by Sato et al. [30], which are based on items featured in the weekly mailer in DH_original, and with a simulated personalization factor in DH_personalized. The third dataset is MovieLens 100K (**ML**) [29], which includes users' ratings on movies and simulated propensity scores based on the ratings and user behaviors. Note that PROPCARE do not require any propensity or exposure data at all. The ground-truth values are only used to evaluate model output. On each dataset, we generate the training/validation/test sets following their original work [30, 29], respectively. We summarize each dataset in Tab. 1, listing the number of users (#users) and items (#items), as well as the average value of several key variables including the observed interaction ($\bar{Y}_{u,i}$), exposure ($\bar{Z}_{u,i}$), causal effect ($\bar{\tau}_{u,i}$) and propensity ($\bar{p}_{u,i}$). Further details can be found in Appendix C.1.

**Baselines**    We compare PROPCARE with the following propensity estimators: (1) **Ground-truth**: Propensity score and exposure values are directly taken from the datasets. (2) **Random**: Propensity scores are assigned randomly between 0 and 1. (3) Item popularity (**POP**): Propensity scores are assigned as item popularity normalized to $(0, 1)$. (4) **CJBPR** [42]: An unbiased recommendation model that optimizes propensity and relevance alternately in a point-wise manner. (5) **EM** [21]: An recommendation model that learns propensity scores in a point-wise manner using an expectation-maximization algorithm.

Note that Ground-truth uses the ground-truth values of propensity $p_{u,i}$ and exposure $Z_{u,i}$ directly as input to train DLCE [30]. All other baselines do not need such ground-truth values in any stage just as PROPCARE. In these methods, the estimated propensity $\hat{p}_{u,i}$ is used to further derive the exposure $\hat{Z}_{u,i}$, in the same way as PROPCARE (see Sect. 4.3). Finally, we utilize the estimated values to train DLCE (see Sect. 4.4).

**Parameter settings**    We tune the hyperparameters based on the validation data, following guidance in the literature. Specifically, in PROPCARE, the trade-off parameter $\lambda$ and $\mu$ are set to 10 and 0.4, respectively, on all datasets. For the downstream causal recommender DLCE, we follow the earlier settings [30]. For other settings and implementation details, refer to Appendix C.2.

**Evaluation metrics**    We evaluate the performance of causality-based recommendation with CP@10, CP@100 and CDCG, whose definitions can be found in Appendix C.3. Additionally, we measure the accuracy of estimated propensity scores w.r.t. the ground-truth values using Kullback-Leibler divergence (KLD) and Kendall's Tau (Tau) [12], and that of estimated exposure using F1 score. Note that all metrics, except KLD, indicate better performance with a larger value.

## 5.2 Results and discussions

We first compare the performance of PROPCARE and the baselines, followed by analyses of model ablation, the regularization term, and various influencing factors. Additional experiments including

Table 2: Performance comparison on downstream causality-based recommendation.

| Methods | DH_original | | | DH_personalized | | | ML | | |
|---|---|---|---|---|---|---|---|---|---|
| | CP@10↑ | CP@100↑ | CDCG↑ | CP@10↑ | CP@100↑ | CDCG↑ | CP@10↑ | CP@100↑ | CDCG↑ |
| Ground-truth | .0658±.001 | .0215±.001 | 1.068±.000 | .1304±.001 | .0445±.001 | 1.469±.003 | .2471±.001 | .1887±.000 | 16.29±.006 |
| Random | .0154±.001 | .0071±.002 | .7390±.004 | .0479±.004 | .0107±.005 | .8316±.039 | .0124±.002 | .0135±.005 | 13.16±.076 |
| POP | .0200±.000 | .0113±.000 | .7877±.001 | .0457±.000 | .0096±.001 | .8491±.002 | -.142±.001 | -.092±.001 | 11.43±.005 |
| CJBPR | .0263±.001 | .0087±.001 | .7769±.002 | .0564±.008 | .0106±.005 | .8528±.032 | -.410±.002 | -.187±.001 | 9.953±.006 |
| EM | .0118±.001 | .0067±.001 | .7247±.001 | .0507±.002 | .0121±.001 | .8779±.003 | -.437±.002 | -.194±.002 | 10.21±.011 |
| PROPCARE | **.0351**±.002 | **.0156**±.001 | **.9268**±.005 | **.1270**±.001 | **.0381**±.000 | **1.426**±.001 | **.0182**±.002 | **.0337**±.002 | **13.80**±.011 |

Results are reported as the average of 5 runs (mean±std). Except Ground-truth, best results are bolded and runners-up are underlined.

Table 3: Performance comparison on propensity score (KLD, Tau) and exposure (F1 score) estimation.

| Methods | DH_original | | | DH_personalized | | | ML | | |
|---|---|---|---|---|---|---|---|---|---|
| | KLD↓ | Tau↑ | F1 score↑ | KLD↓ | Tau↑ | F1 score↑ | KLD↓ | Tau↑ | F1 score↑ |
| Random | .5141±.001 | .0002±.000 | .4524±.013 | 3.008±.002 | .0001±.000 | .4463±.021 | .0363±.002 | .0002±.000 | .4511±.022 |
| POP | .5430±.000 | **.4726**±.000 | .2851±.000 | 4.728±.000 | **.6646**±.000 | .2772±.000 | .0615±.000 | **.4979**±.000 | .5050±.000 |
| CJBPR | .3987±.008 | .3279±.011 | .2853±.005 | 2.650±.022 | .6477±.013 | .2825±.005 | .0230±.006 | .4956±.045 | **.5189**±.020 |
| EM | .6380±.002 | .0834±.000 | .4974±.001 | 2.385±.001 | .0934±.002 | .4954±.009 | .0517±.001 | .1321±.002 | .3653±.005 |
| PROPCARE | **.3851**±.023 | .3331±.065 | **.5846**±.006 | **1.732**±.038 | .4706±.072 | **.6059**±.017 | **.0204**±.005 | .3889±.034 | .4847±.020 |

Results are styled in the same way as in Tab. 2.

comparison to conventional recommendation methods, evaluation on an alternative backbone, and a scalability study are presented in Appendix D.

**Performance comparison** We evaluate PROPCARE against the baselines in two aspects: (1) The downstream causality-based recommendation using the estimated propensity and exposure; (2) The accuracy of the estimated propensity and exposure.

We first illustrate the performance of causality-based recommendation in Tab. 2. It is not surprising that Ground-truth achieves the best causal effect by incorporating actual propensity and exposure values in DLCE. However, since ground-truth values are often unavailable, we rely on estimations. Among all baselines, PROPCARE most closely approaches Ground-truth's performance. Notably, in the DH_personalized dataset, PROPCARE exhibits only a 6.6% average decrease from Ground-truth across three metrics, significantly outperforming the second-best EM which suffers a 56.6% drop. Furthermore, PROPCARE surpasses the point-wise CJBPR and EM, implying the advantage of our pairwise formulation based on Assumption 1.

Next, we analyze the accuracy of propensity and exposure estimation in Tab. 3. Among the baselines, POP performs the best in Kendall's Tau. However, the causality metrics of POP is poor (see Tab. 2) due to the ill-fit propensity distribution, reflected in its large KLD from the ground-truth distribution. The estimation of exposure is also challenging for POP in most cases. In contrast, PROPCARE demonstrates outstanding performance in F1 score and KLD, leading to effective causal metrics. Although its Tau scores lag behind some baselines, a robust distribution on propensity and accurate binary predictions of exposure still contribute to good causal performance. The results in Tab. 3 highlight that causality-based recommendation is influenced by multiple factors, rather than relying solely on a single aspect of estimation. We will discuss these influencing factors further toward the end of this part.

**Ablation study** To evaluate the impact of our key design motivated by Assumption 1, we derive five variants from Eq. (6): (1) **NO_P**: removing the constraint on estimated $\hat{p}_{u,i}$ by deleting the term with $f_p(\mathbf{x}_{u,i}) - f_p(\mathbf{x}_{u,j})$; (2) **NO_R**: removing the constraint on estimated $\hat{r}_{u,i}$ by deleting the term with $f_r(\mathbf{x}_{u,j}) - f_r(\mathbf{x}_{u,i})$; (3) **NO_P_R**: removing $\mathcal{L}_{\text{pop}}$ entirely from the overall loss to eliminate Assumption 1 altogether; (4) **NEG**: reversing Assumption 1 by replacing $\text{Sgn}_{i,j}$ with $-\text{Sgn}_{i,j}$ to assume that more popular items have smaller propensity scores; (5) $\kappa = 1$: setting all $\kappa_{u,i,j}$'s to a constant 1, resulting in equal weighting of all training triplets. Their causal performances are illustrated in Fig. 3. Comparing to the full version of PROPCARE, NO_R and NO_P show a small drop in performance due to the absence of additional constraints on propensity or relevance, indicating that the pairwise loss is still partially effective. The drop in $\kappa = 1$ highlights the need for controlling the similarity between interaction probabilities. The further drop observed in NO_P_R

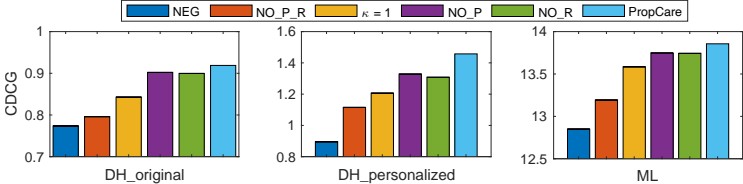

Figure 3: Ablation study on PROPCARE.

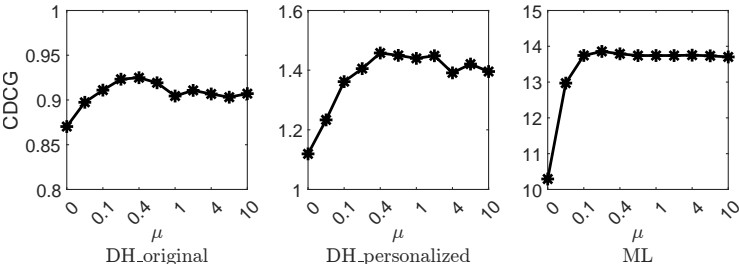

Figure 4: Effect of regularization term.

after removing the entire assumption further demonstrates the validity of our assumption and the effectiveness of $\mathcal{L}$pop. The worst performance of NEG indicates that contradicting Assumption 1 greatly affects the results of causality-based recommendation.

**Effect of regularization**   We examine how effective the regularization term is by varying the trade-off hyperparameter $\mu$, as used in Eq. (7), over the set $\{0, 0.01, 0.1, 0.2, 0.4, 0.8, 1, 2, 4, 8, 10\}$. The results, shown in Fig. 4, indicate that when $\mu$ is set to 0, which means the regularization term is completely removed, the causal performance is significantly compromised. This demonstrates the advantage of regularizing the propensity with a beta distribution during the training of PROPCARE. As $\mu$ increases, the causal performance gradually improves and reaches its peak within the range of $[0.2, 0.8]$. We suggest further exploration by fine-tuning the hyperparameter within this interval.

**Factors influencing causality-based recommendation**   We further investigate the factors crucial to causality-based recommendation, by injecting noises into ground-truth propensity or exposure values. In Fig. 5(a), we randomly flip a fraction of the exposure values while using the ground-truth propensity scores for DLCE training. The flipping ratios range from $\{0, 0.01, 0.05, 0.1, 0.15, 0.2, 0.3, 0.4, 0.5\}$. In Fig. 5(b), we add Gaussian noises to the propensity scores, where the variances of the noise range from $\{0, 0.1, \ldots, 0.5\}$ while using the ground-truth exposure. The results in Fig. 5 consistently show a decrease in performance due to misspecified exposure or propensity. To further examine the correlation between estimation accuracy and recommendation performance, we create scatter plots in Fig. 6 for the DH_original dataset. Each data point represents a baseline labeled with its name. The plots reveal a general trend where the CDCG is correlated with the accuracy of estimations.

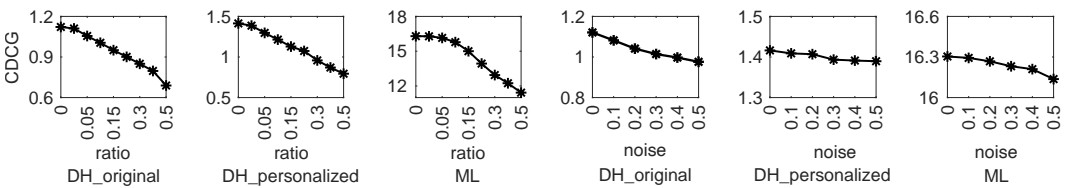

(a) Ground-truth propensity score + randomly flipped exposure    (b) Noisy propensity score + ground-truth exposure

Figure 5: Analysis of factors that influence causality-based recommendation performance.

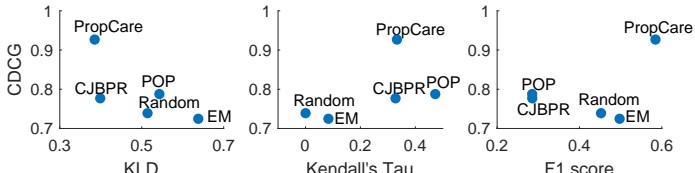

Figure 6: Correlation analysis on factors influencing recommendation.

Table 4: Case study of an anonymous user.

| Ground-truth | POP | CJBPR | PROPCARE |
|---|---|---|---|
| garlic bread (1176) | bananas (1310) | ‡ **fluid milk** (1169) | ‡ **infant soy** (1232) |
| ‡ **cleansing wipes** (737) | toilet tissue (742) | bananas (1310) | ‡ **fluid milk** (1169) |
| ‡ **fluid milk** (1169) | ‡ **fluid milk** (1169) | cereal (1090) | bananas (1310) |
| ‡ **primal** (807) | white bread (675) | **strawberries** (834) | pure juice (1277) |
| alkaline batteries (754) | ¶ tortilla chips (634) | margarine tubs/bowls (1245) | coffee creamers (1169) |

Each column represents the recommendation list output by DLCE trained with the estimated propensity and exposure by the corresponding baseline. The purchased items are highlighted in bold. Items with positive causal effect ($\tau_{u,i} = 1$) and negative causal effect ($\tau_{u,i} = -1$) are marked by ‡ and ¶, respectively, and unmarked items have zero causal effect ($\tau_{u,i} = 0$). Numbers in brackets are the popularity ranks in the training set.

### 5.3 Case study

We conduct a case study to demonstrate the advantages of PROPCARE in a practical ranking-based recommendation scenario. In Tab. 4, we analyze the top-5 recommended items for an anonymous user with ID 2308 in the DH_personalized dataset. In the first column, by utilizing ground-truth propensity scores and exposure, DLCE effectively generates a ranking list where most items have a positive causal effect. All items with a positive causal effect were eventually purchased, achieving the goal of causality-based recommendation. Comparing the lists generated by CJBPR and PROPCARE, it is evident that the associated causal effects of the purchased items differ. For example, in the CJBPR list, recommending "strawberries" has zero causal effect, indicating that the user could have still purchased it even without recommendation. In contrast, PROPCARE recommends "infant soy", which has a positive causal effect, making it a more ideal choice. Overall, given the list recommended by CJBPR, the user would only purchase "strawberries" and "fluid milk". However, given the list from PROPCARE, in addition to "infant soy" and "fluid milk", the user may still purchase "strawberries" even without being recommended due to its zero causal effect. Besides, POP tends to recommend popular items but with a lower causal effect, even including an item with a negative causal effect. The results suggest that POP is not an appropriate tool for estimating propensity scores in the context of causality-based recommendation.

## 6 Conclusion

In this paper, we introduced PROPCARE, a propensity estimation model for causality-based recommendation systems without the need to access ground-truth propensity and exposure data. Leveraging our observation on the pairwise characteristics between propensity scores and item popularity, we formulated a key assumption and incorporated it as prior information to enhance our estimation, thereby improving causality-based recommendation. A theoretical analysis was presented to understand the factors influencing the bias in estimated causal effects, thereby informing model design and evaluation. Empirical studies demonstrated the superiority of PROPCARE over the baselines. Future research avenues include exploring direct exposure estimation without propensity scores, and investigating parametric causal effect estimators that are potentially more powerful.

## Acknowledgments

This research is supported by the Agency for Science, Technology and Research (A*STAR) under its AME Programmatic Funds (Grant No. A20H6b0151). Any opinions, findings and conclusions or recommendations expressed in this material are those of the author(s) and do not reflect the views of the A*STAR. Dr. Yuan Fang also acknowledges the Lee Kong Chian Fellowship awarded by Singapore Management University for the support of this work.

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

# Appendices

## A   Training procedure of PROPCARE

We present the pseudocode for training our proposed PROPCARE in Algorithm 1. The training steps involve calculating various loss terms including $\mathcal{L}_{\text{naïve}}$, $\mathcal{L}_{\text{pop}}$ and the regularization term. We update all learnable parameters based on the total loss defined in Eq. (7) of the main text.

---

**Algorithm 1:** Training PROPCARE

---

**Input:** Observed training interaction data $D$.
**Output:** Model parameters $\Theta$.

Initialize model parameters $\Theta$;
**while** *not converged* **do**
    **foreach** *user-item pair* $(u, i)$ *in* $D$ **do**
        Compute $\mathcal{L}_{\text{naïve}}$ by Eq. (3) of the main text;
        Sample an item $j$ for each pair $(u, i)$ from $\{1, 2, \ldots, I\}\backslash i$;
        Compute $\mathcal{L}_{\text{pop}}$ by Eq. (6) of the main text;
        Compute the total loss in Eq. (7) of the main text;
    **end**
    Update $\Theta$ by backpropagation of the loss in Eq. (7) of the main text;
**end**
**return** $\Theta$.

---

We then analyze the time complexity of the training procedure. We first consider the computation of $\mathcal{L}_{\text{naïve}}$. This involves three MLP models: $f_e$, $f_p$, and $f_r$. As we have to compute $\mathbf{x}_{u,i}$, $\hat{p}_{u,i}$ and $\hat{r}_{u,i}$ for each user-item pair, each MLP incurs a time complexity of $\mathcal{O}(n)$, where $n = |D|$, the number of user-item pairs in the training data. To further find the interactions $\{(y_{u,i})\}$, we need to compute the product between $\hat{p}_{u,i}$ and $\hat{r}_{u,i}$, which also has a linear time complexity of $\mathcal{O}(n)$. Next, to compute $\mathcal{L}_{\text{pop}}$, we need to sample another item $j$ for each user-item pair in the training data, which has a time complexity of $\mathcal{O}(n)$. For each tuple $(u, i, j)$, we need to further compute $\hat{p}_{u,j}$ and $\hat{r}_{u,j}$, each taking $\mathcal{O}(n)$ time. Finally, the time complexity to compute the regularization term is also $\mathcal{O}(n)$, as the empirical distribution is based on all estimated propensity scores for training pairs. In summary, a series of $O(n)$ procedures are carried out for the user-item pairs in the training set $D$, resulting in an overall linear time complexity of $O(n)$. This demonstrates the scalability of our proposed PROPCARE, which we further analyze empirically in Appendix D.2.

## B   Bias of the estimated causal effect

Here, we present the the proof of Proposition 1, which have appeared in the main text, Sect. 4.5.

**Proof 1** *As defined earlier [30], we can model the potential outcomes as follows,*

$$\hat{Y}^1_{u,i} = \frac{Z_{u,i}Y_{u,i}}{p_{u,i}}, \tag{a.1}$$

$$\hat{Y}^0_{u,i} = \frac{(1 - Z_{u,i})Y_{u,i}}{1 - p_{u,i}}. \tag{a.2}$$

*Note that in Eqs. (a.1) and (a.2), the $Y_{u,i}$'s on the right-hand side can be replaced with $Y^1_{u,i}$ and $Y^0_{u,i}$, respectively. This substitution is valid because when $Z_{u,i} = 1$, the $Y_{u,i}$ term in Eq. (a.1) is equivalent to $Y^1_{u,i}$ by definition. In this scenario, $\hat{Y}^0_{u,i}$ is always 0, regardless of the value of $Y^0_{u,i}$. Similarly, when $Z_{u,i} = 0$, the $Y_{u,i}$ term in Eq. (a.2) is equivalent to $Y^0_{u,i}$. Therefore, the IPS estimator for the causal effect, in Eq. (1) of the main text, can be rewritten as*

$$\hat{\tau}_{u,i} = \frac{Z_{u,i}Y^1_{u,i}}{p_{u,i}} - \frac{(1 - Z_{u,i})Y^0_{u,i}}{1 - p_{u,i}}. \tag{a.3}$$

*As established earlier [30], if the propensity score $p_{u,i}$ and exposure $Z_{u,i}$ are correctly assigned, the expectation of the estimated causal effect from the IPS estimator is*

$$\mathbb{E}\left[\hat{\tau}_{u,i}\right] = Y_{u,i}^1 - Y_{u,i}^0 = \tau_{u,i}. \tag{a.4}$$

*That is to say, if we have ground-truth propensity score as well as exposure, the estimated causal effect is unbiased. If we have only ground-truth exposure but have to estimate propensity scores by substituting $p_{u,i}$ with $\hat{p}_{u,i}$ in Eq. (a.3), denote the resulting estimator for the casual effect as $\hat{\tau}_{u,i}'$. Then, the expectation and bias of $\hat{\tau}_{u,i}'$ are*

$$\mathbb{E}\left[\hat{\tau}_{u,i}'\right] = \frac{p_{u,i}Y_{u,i}^1}{\hat{p}_{u,i}} - \frac{(1-p_{u,i})Y_{u,i}^0}{1-\hat{p}_{u,i}}, \tag{a.5}$$

$$\text{Bias}(\hat{\tau}_{u,i}') = \mathbb{E}\left[\hat{\tau}_{u,i}'\right] - \tau_{u,i} = \left(\frac{p_{u,i}}{\hat{p}_{u,i}} - 1\right)Y_{u,i}^1 - \left(\frac{1-p_{u,i}}{1-\hat{p}_{u,i}} - 1\right)Y_{u,i}^0. \tag{a.6}$$

*Finally, in our setup, both $Z_{u,i}$ and $p_{u,i}$ are estimated. Hence, denote the resulting estimator based on $\hat{Z}_{u,i}$ and $\hat{p}_{u,i}$ as $\hat{\tau}_{u,i}''$. We can obtain its bias relative to $\hat{\tau}_{u,i}'$ as follows.*

$$\text{Bias}(\hat{\tau}_{u,i}'') - \text{Bias}(\hat{\tau}_{u,i}') = \mathbb{E}\left[\hat{\tau}_{u,i}''\right] - \mathbb{E}\left[\hat{\tau}_{u,i}'\right]$$

$$= \mathbb{E}\left[\frac{\hat{Z}_{u,i}Y_{u,i}^1}{\hat{p}_{u,i}} - \frac{(1-\hat{Z}_{u,i})Y_{u,i}^0}{1-\hat{p}_{u,i}} - \frac{Z_{u,i}Y_{u,i}^1}{\hat{p}_{u,i}} + \frac{(1-Z_{u,i})Y_{u,i}^0}{1-\hat{p}_{u,i}}\right]$$

$$= \mathbb{E}\left[\frac{(\hat{Z}_{u,i} - Z_{u,i})Y_{u,i}^1}{\hat{p}_{u,i}} - \frac{(Z_{u,i} - \hat{Z}_{u,i})Y_{u,i}^0}{1-\hat{p}_{u,i}}\right]$$

$$= \frac{\mathbb{E}\left[\hat{Z}_{u,i} - Z_{u,i}\right]}{\hat{p}_{u,i}}Y_{u,i}^1 + \frac{\mathbb{E}\left[\hat{Z}_{u,i} - Z_{u,i}\right]}{1-\hat{p}_{u,i}}Y_{u,i}^0. \tag{a.7}$$

*By adding Eqs. (a.6) and (a.7), we are able to obtain the bias of $\hat{\tau}_{u,i}''$*

$$\text{Bias}(\hat{\tau}_{u,i}'') = \text{Bias}(\hat{\tau}_{u,i}') + \text{Bias}(\hat{\tau}_{u,i}'') - \text{Bias}(\hat{\tau}_{u,i}')$$

$$= \left(\frac{p_{u,i} + \mathbb{E}\left[\hat{Z}_{u,i} - Z_{u,i}\right]}{\hat{p}_{u,i}} - 1\right)Y_{u,i}^1 - \left(\frac{1 - p_{u,i} - \mathbb{E}\left[\hat{Z}_{u,i} - Z_{u,i}\right]}{1-\hat{p}_{u,i}} - 1\right)Y_{u,i}^0. \tag{a.8}$$

*This concludes the proof of Proposition 1.* □

## C   Additional experimental settings

We describe more details on the datasets, implementation and evaluation metrics.

### C.1   Descriptions of datasets

We introduce additional details on data generation and splitting.

**Data processing and generation**   We perform data processing and generation steps per the earlier studies on the DunnHumby[2] (DH) [30] and MovieLens[3] [29] (ML) datasets.

Specifically, for DH, the items that appear in the weekly mailer are deemed as recommended (i.e., exposed) items. The empirical distribution of $Y_{u,i}^1$ can be found by tallying the weeks in which item $i$

---

[2]The raw data are available at https://www.dunnhumby.com/careers/engineering/sourcefiles.
[3]The raw data are available at https://grouplens.org/datasets/movielens.

was both recommended to and purchased by user $u$ (or purchased but *not* recommended when dealing with $Y^0_{u,i}$). The ground-truth values of $Y^1_{u,i}$ and $Y^0_{u,i}$ are then sampled from their respective empirical distributions, which are used to calculate the ground-truth causal effect, as follows.

$$\tau_{u,i} = Y^1_{u,i} - Y^0_{u,i}. \tag{a.9}$$

Subsequently, two different ways of simulating the ground-truth propensity scores have been attempted. In DH_original, the propensity score $p_{u,i}$ is defined based on the number of weeks in which the item was recommended while the user visited the retailer during the same week. For DH_personalized, the propensity score $p_{u,i}$ is established based on the position of item $i$ in a simulated ranking based on user $u$'s probability of interaction with $i$. The ground-truth exposure $Z_{u,i}$ is then sampled from a Bernoulli distribution whose parameter is set to the propensity score $p_{u,i}$. Finally, the ground-truth interaction can be computed as follows.

$$Y_{u,i} = Z_{u,i} Y^1_{u,i} + (1 - Z_{u,i}) Y^0_{u,i}. \tag{a.10}$$

For ML, the empirical distributions of $Y^1_{u,i}$ and $Y^0_{u,i}$ are derived from the interaction log using matrix factorization-based techniques. The propensity score is determined by a simulated ranking for each user, similar to DH_personalized. Subsequently, the ground-truth values of the exposure, causal effect and interaction is sampled and established following the same process in DH.

**Data splitting** The above data generation steps are applied to both DH and ML datasets to generate training, validation and testing sets. For the DH datasets, the data generation process is repeated 10 times to simulate the 10-week training data, once more to simulate the 1-week validation data, and 10 more times to simulate the 10-week testing data. For the ML dataset, the generation is repeated once each to generate the training, validation, and testing data, respectively.

It is worth noting that the data generation and splitting processes are dependent on some form of simulation. We utilize such "semi-simulated" data for a number of reasons. First, the true causal effects are not observable due to their counterfactual nature. Second, ground-truth propensity scores and exposure are often not available in public datasets, which nonetheless are essential for model evaluation (even though they are not required for our model training). Third, while some datasets [20, 36, 6] do include information on item impressions or exposure statuses, they often provide a one-sided view of the situation. Specifically, the vast majority, if not all, of the item interactions are preceded by prior impressions. This leaves limited scope for investigating item interactions that occur without prior impressions.

### C.2 Implementation details

Let us first present key implementation choices regarding propensity and exposure modeling in PROPCARE. On all datasets, we set $\alpha = 0.2$ and $\beta = 1.0$ for the Beta distribution used in the regularization term. To derive $\hat{Z}_{u,i}$ based on $\hat{p}_{u,i}$, we implement Norm as a Z-score normalization function, such that $\hat{Z}_{u,i} = 1$ if $\text{Norm}(\hat{p}_{u,i}) \geq \epsilon$, where the threshold $\epsilon$ is set to 0.2 for DH_original and DH_personalized, and 0.15 for ML. To address the bias introduced by $\hat{p}_{u,i}$, we employ a practical trick to scale the estimated propensity by a constant factor $c$, i.e., $\hat{p}_{u,i} \leftarrow c \times \hat{p}_{u,i}$, prior to training DLCE. This technique aims to further minimize the disparity in the scale between $p_{u,i}$ and $\hat{p}_{u,i}$ in order to improve the accuracy of causal effect estimation, according to the theoretical analysis in Sect. 4.5. For DH_original and DH_personalized, the scaling factor $c$ is set to 0.8, while for ML it is set to 0.2. The hyperparameters $\epsilon$ and $c$ are tuned using validation data. In particular, we perform a grid search where $\epsilon$ is searched over the range (0,1) in steps of 0.05 and $c$ is searched over the range (0,1] in steps of 0.1.

With regard to the model architecture for PROPCARE, we randomly initialize $\mathbf{x}_u$ and $\mathbf{x}_i \in \mathbb{R}^{128}$ as user and item ID features. The embedding model $f_e$ takes $(\mathbf{x}_u \| \mathbf{x}_i)$ as input and is implemented as an MLP with 256, 128 and 64 neurons for its layers. $f_p$ and $f_r$ are both implemented as MLPs with 64, 32, 16, 8 neurons for the hidden layers and an output layer activated by the sigmoid function. Except the output layer, all layers of $f_e$ and hidden layers of $f_p$ and $f_r$ are activated by the LeakyReLU function. Finally, PROPCARE is trained with a stochastic gradient descent optimizer using mini-batches, with a batch size set to 5096.

Table a.1: Performance comparison with conventional interaction-based recommenders.

| Methods | DH_original | | | DH_personalized | | | ML | | |
|---|---|---|---|---|---|---|---|---|---|
| | CP@10↑ | CP@100↑ | CDCG↑ | CP@10↑ | CP@100↑ | CDCG↑ | CP@10↑ | CP@100↑ | CDCG↑ |
| MF | .0206±.001 | .0118±.000 | .8569±.002 | .0433±.001 | .0196±.000 | .9699±.002 | -.460±.004 | -.205±.002 | 9.421±.023 |
| BPR | .0301±.000 | .0138±.000 | .8983±.002 | .0476±.001 | .0245±.000 | 1.018±.001 | -.408±.002 | -.197±.001 | 9.898±.028 |
| LightGCN | .0309±.001 | .0149±.001 | .9113±.004 | .0821±.001 | .0263±.001 | 1.095±.002 | -.342±.006 | -.177±.002 | 10.16±.050 |
| PROPCARE | **.0351**±.002 | **.0156**±.001 | **.9268**±.005 | **.1270**±.001 | **.0381**±.000 | **1.426**±.001 | **.0182**±.002 | **.0337**±.002 | **13.80**±.011 |

Results are reported as the average of 5 runs (mean±std). Best results are bolded.

For the baseline CJBPR, we have used its authors' implementation[4] and modified its mini-batch setting and optimizer to be identical to PROPCARE. For the baseline EM, we have implemented their proposed propensity model in Python. To conduct downstream causality-based recommendation, we have used the authors' implementation of DLCE[5] following the settings in their work [30]. For a fair comparison, the normalization function Norm and constant scaling factor $c$ are identically applied to all baselines, which generally improve the performance across the board.

We implement PROPCARE using TensorFlow 2.11 in Python 3.10. All experiments were conducted on a Linux server with a AMD EPYC 7742 64-Core CPU, 512 GB DDR4 memory and four RTX 3090 GPUs.

### C.3 Evaluation metrics

Unlike commonly used metrics like precision and NDCG that reward all positive interactions regardless of whether the item is exposed or not, we use a variant of them that only reward positive interactions resulting from item exposure. Concretely, we use the Causal effect-based Precision (CP) and Discounted Cumulative Gain (CDCG) as defined in existing work [30].

$$\text{CP@}K = \frac{1}{U} \sum_{u=1}^{U} \sum_{i=1}^{I} \frac{\mathbf{1}(\text{rank}_u(\hat{s}_{u,i}) \leq K)\tau_{u,i}}{K}, \tag{a.11}$$

$$\text{CDCG} = \frac{1}{U} \sum_{u=1}^{U} \sum_{i=1}^{I} \frac{\tau_{u,i}}{\log_2 \left(1 + \text{rank}_u(\hat{s}_{u,i})\right)}, \tag{a.12}$$

where $\mathbf{1}(\cdot)$ is an indicator function, and $\text{rank}_u(\hat{s}_{u,i})$ returns the position of item $i$ in the ranking list for user $u$ as determined by the ranking score $\hat{s}_{u,i}$. In our paper, we report CP@10, CP@100 and CDCG. Note that since $\tau_{u,i}$ can be $-1$, these metrics can be negative.

## D  Additional experiments

We present additional empirical results on the comparison to conventional recommender systems, as well as the robustness and scalability of PROPCARE.

### D.1  Comparison to interaction-based recommenders

To demonstrate the advantage of PROPCARE over conventional interaction-based recommender systems, we compare with three well-known models, namely Matrix Factorization[6] (MF) [13], Bayesian Personalized Ranking[6] (BPR) [23] and LightGCN[7] [7]. Note that we use only interaction data $Y_{u,i}$ as training labels for these methods and rank the items for each user based on the predicted $\hat{y}_{u,i}$, as in their classical paradigm. We evaluate the causal performance with CP@10, CP@100 and CDCG in Tab. a.1. From the results we can observe that the conventional interaction-based methods do not demonstrate strong causal performance.

A fundamental reason is that the conventional models indiscriminately reward all positive interactions, regardless of whether the candidate items have been exposed or not. To delve deeper into the impact

---

[4]Available at `https://github.com/Zziwei/Unbiased-Propensity-and-Recommendation`.

[5]Available in ancillary files at `https://arxiv.org/abs/2008.04563`.

[6]Implementation available in ancillary files at `https://arxiv.org/abs/2008.04563`.

[7]Implementation available at `https://github.com/kuandeng/LightGCN`.

Table a.2: Influence of exposure status on causal effect.

|  | $\mathbb{E}[\tau_{u,i}|Y_{u,i}=1, Z_{u,i}=1]$ | $\mathbb{E}[\tau_{u,i}|Y_{u,i}=1, Z_{u,i}=0]$ |
|---|---|---|
| DH_original | .8882 | -.901 |
| DH_personalized | .8882 | -.985 |
| ML | .8623 | -.818 |

Table a.3: Employing CausE as the causality-based recommendation backbone.

| Methods | DH_original | | | DH_personalized | | | ML | | |
|---|---|---|---|---|---|---|---|---|---|
|  | CP@10↑ | CP@100↑ | CDCG↑ | CP@10↑ | CP@100↑ | CDCG↑ | CP@10↑ | CP@100↑ | CDCG↑ |
| Ground-truth | .0495±.002 | .0216±.002 | 1.020±.004 | .0663±.003 | .0241±.002 | 1.105±.003 | .1849±.002 | .1438±.004 | 15.25±.007 |
| POP | .0059±.001 | .0097±.002 | .7759±.002 | .0427±.002 | .0159±.001 | .9616±.002 | -.191±.001 | -.036±002 | 12.27±.006 |
| CJBPR | .0073±.001 | .0100±.005 | .7809±.003 | .0451±.002 | .0165±.003 | .9621±.005 | -.217±.001 | -.044±.001 | 12.05±.006 |
| EM | .0065±.000 | .0103±.001 | .7802±.002 | .0478±.001 | .0166±.001 | .9819±.006 | -.197±.002 | -.041±.002 | 12.24±.009 |
| PROPCARE | .0123±.001 | .0114±.001 | .8084±.001 | .0580±.005 | .0201±.001 | 1.052 ±.002 | -.138±.001 | -.035±.003 | 12.40±.009 |

Results are reported as the average of 5 runs (mean±std). Except Ground-truth, best results are bolded and runners-up are underlined.

of different exposure statuses on the causal effect of user-item pairs with positive interactions, we calculate $\mathbb{E}[\tau_{u,i}|Y_{u,i}=1, Z_{u,i}=1]$ and $\mathbb{E}[\tau_{u,i}|Y_{u,i}=1, Z_{u,i}=0]$ for each dataset, as presented in Tab. a.2. The divergent results given different exposure statuses imply that exposure plays a significant role in the causal effect involving positive interactions. Conventional interaction-based models ignore exposure and may rank items in a way that negatively impacts the causal effect. Between the two DH datasets, DH_personalized exhibits a lower value of $\mathbb{E}[\tau_{u,i}|Y_{u,i}=1, Z_{u,i}=0]$, which explains the greater causal performance gap between the conventional models and PROPCARE than the gap on DH_original. This greater gap arises because recommending items with $Y_{u,i}=1, Z_{u,i}=0$ lowers the causal effect more on DH_personalized than on DH_original.

## D.2 Robustness to alternative backbone and scalability analysis

To show the robustness of PROPCARE, we opt for CausE[6] [2], another causality-based recommender, as an alternative backbone to DLCE. Like DLCE, CausE makes causality-based recommendation given the estimated propensity and exposure data from PROPCARE or the baselines. The results in Tab. a.3 show a similar pattern as using DLCE in the main text. That is, our PROPCARE consistently outperforms other baselines even with a different causality-based model as the backbone.

We further examine the scalability of our proposed PROPCARE on increasingly larger datasets, namely, MovieLens 1M[8], MovieLens 10M[8] and MovieLens 20M[8]. In Tab a.4, we report the training times of both PROPCARE and DLCE, in hours. We observe that the training of PROPCARE generally follows a linear growth, in line with the time complexity anlysis in Appendix A. Moreover, PROPCARE only presents a marginal overhead on top of the backbone DLCE, showing its feasibility in working with existing backbones.

Table a.4: Training times for PROPCARE and DLCE, in hours.

|  | MovieLens 1M | MovieLens 10M | MovieLens 20M |
|---|---|---|---|
| PROPCARE | .0814 | 1.494 | 3.170 |
| DLCE | 2.1759 | 22.658 | 40.658 |

---

[8]Available at `https://grouplens.org/datasets/movielens/`.

