# OpenReview forum: "Estimating Propensity for Causality-based Recommendation without Exposure Data"
_NeurIPS.cc/2023/Conference — NeurIPS 2023 poster_

### Official Review · Reviewer_5c9m · 2023-06-19

**Soundness:** 3 good
**Presentation:** 4 excellent
**Contribution:** 3 good
**Rating:** 7
**Confidence:** 5

**Summary:**

This paper presents a review of a novel method proposed for estimating causal effects in situations where no observation of the treatment variable is available. The authors introduce an innovative approach that utilizes interactions to approximate missing exposure or propensity data. The method relies on several key assumptions, including the significance of "popularity" as a strong confounder of both treatment and outcome variables, as well as the proposition that the outcome variable is a product of the unobserved propensity and relevance (which is a function of historical popularity). Additionally, the paper assumes a monotonic relationship between popularity and exposure (Assumption 1) and utilizes a parametric assumption based on the Beta distribution.

The paper does a great job reviewing existing literature. And the simulation section provides a thorough comparison with baselines and oracles that have access to the treatment variable. The ablation study of its own method is also very comprehensive which provided intuition on the most important factors of the proposed model.

**Strengths:**

The proposed method is very novel by leveraging strong intuitions in the domain to find strong confounders that can reliably impute the exposure variable through a learnable objective. The paper presents a fresh new perspective into the causal inference application to Recommendation systems when privacy limits the access to treatment assignments in production applications.


**Weaknesses:**

N/A

**Questions:**

Question 1: PropCare is a solid method when all the assumptions are met: The method assumes `popularity` as a strong confounder of treatment and of Y.
I wonder how general the assumptions are when the treatment changes or when applied to other datasets when the treatment does not have a single strong-enough confounder.

Question 2: about line 291 (simulation section): Can authors share the distribution of the fitted propensity scores using all the compared methods? Wonder if POP's bad performance is due to a practical violation of the positivity assumption, plus not mitigating it through bounding the fitted value between e.g. [0.02, 0.98]. Since the PropCare method handles the positivity assumption violation in its regularization term, it should be a fair comparison to perform comparable mitigation for the other baselines.

---

> ### Author Rebuttal · Authors · 2023-08-08
>
> Thanks for acknowledging our novelty and literature review. we will give answers to each question asked by this reviewer below.
>
> **Q1:** PropCare is a solid method when all the assumptions are met: The method assumes popularity as a strong confounder of treatment and of Y. I wonder how general the assumptions are when the treatment changes or when applied to other datasets when the treatment does not have a single strong-enough confounder.
>
> **A1:** Assuming the casual relationship between popularity and treatment is quite general in many previous works. For example, [a,b] explicitly state the confounder role of popularity in the causal graph, [c,d] also empirically investigate how popularity affects the item exposure or predicted ratings. Hence, we believe the assumptions are reasonably general in most of the situations.
>
> **Q2:** about line 291 (simulation section): Can authors share the distribution of the fitted propensity scores using all the compared methods? Wonder if POP's bad performance is due to a practical violation of the positivity assumption, plus not mitigating it through bounding the fitted value between e.g. [0.02, 0.98]. Since the PropCare method handles the positivity assumption violation in its regularization term, it should be a fair comparison to perform comparable mitigation for the other baselines.
>
> **A2:** We show the distribution of fitted propensity scores by baselines for DH\_original dataset in **Figure 1 of global rebuttal PDF**. From the results it can be seen that the long tailed distribution of POP where most of the values are close to 0 might be a reason for its bad performance, as the violation of positivity assumption may lead to large variance in the outputs of causal recommendation [e,f]. However, to avoid that, our backbone model DLCE [e] has already deployed a **capping threshold** $\chi$ to reduce variance of the prediction. Besides, in data generation steps of [e], the simulated propensity are **clipped** in the range of [$10^{-6}$, $1-10^{-6}$] as a mitigation. In our experiment, we follow the same clipping step after we obtained the estimated propensity of each baseline by default. Hence, *give these mitigation steps, it is a fair comparison*.
>
> Additionally, please note that we have also done an ablation study and parameter analysis of the regularization term in **Appendix D.1**, and we found that even without the regularizer (i.e., $\mu$=0), PropCare still outperformed POP, e.g., 0.87 vs. 0.79 in CDCG on DH_original. This implies that the better performance of PropCare is not just due to the regularizer.
>
> **References**
>
> [a] Zhang, Yang, et al. "Causal intervention for leveraging popularity bias in recommendation." SIGIR 2021.
>
> [b] Tianxin Wei, et al. "Model-Agnostic Counterfactual Reasoning for Eliminating Popularity Bias in Recommender System." KDD. 2021
>
> [c] Zhongzhou Liu, et al. "Mitigating Popularity Bias for Users and Items with Fairness-centric Adaptive Recommendation." ACM Trans. Inf. Syst. 41, 3, Article 55. 2023
>
> [d] Ziwei Zhu, et al. "Measuring and Mitigating Item Under-Recommendation Bias in Personalized Ranking Systems." SIGIR. 2020.
>
> [e] Masahiro Sato, et al. 2020. ”Unbiased Learning for the Causal Effect of Recommendation”. RecSys. 2020.
>
> [f] Teng Xiao and Suhang Wang."Towards Unbiased and Robust Causal Ranking for Recommender Systems". WSDM. 2022.

---

> > ### Comment · Reviewer_5c9m · 2023-08-14
> >
> > I thank the authors for the response and doing additional experimentation to address my concerns. After reading the rebuttal and the comments from other reviewers, I increase my score from Weak Accept to Accept.

---

> > > ### Author Response · Authors · 2023-08-14
> > >
> > > Dear Reviewer 5c9m,
> > >
> > > We sincerely thank you for your prompt and timely response to our rebuttal.

---

### Official Review · Reviewer_Pdiw · 2023-07-02

**Soundness:** 2 fair
**Presentation:** 2 fair
**Contribution:** 2 fair
**Rating:** 3
**Confidence:** 5

**Summary:**

This paper proposes a propensity estimation model for causality-based recommendation without accessing the ground-truth propensity score or exposure data. Prior knowledge about item popularity is utilized to estimate the propensity score. A theoretical analysis is provided to understand the proposed model.

**Strengths:**

- This paper investigates an interesting problem in causality-based recommendation, propensity score estimation, and proposes a model to get rid of the requirements for ground-truth exposure data.
- A theoretical analysis is provided to understand the critical factors of the proposed model.
- Ablation study is conducted.

**Weaknesses:**

- I have serious concerns about the evaluation framework in this paper. All the recommendation experiments are based on one single DLCE model, and the adopted baselines for comparison are rather weak. I suggest the authors to include more advanced baselines, and more importantly, to compare with a wider range of backbone models. Just to list a few widely acknowledged causal recommendation approaches [1-3] which all do not require exposure data.
- The technical contributions of the paper is limited. The proposed pairwise relationship between item popularity and propensity score is similar to the design in [1,2], which also leverage popularity to serve as a *soft* proxy for propensity score.
- The adopted three datasets are very small, considering modern recommendation platforms.
- In Table 3, the proposed PropCare model does not achieve the best performance with respect to Tau and F1 score, outperformed by both POP and CJBPR.
- The case study is hard to follow and not convincing to me.

[1] Bonner, Stephen, and Flavian Vasile. "Causal embeddings for recommendation." Proceedings of the 12th ACM conference on recommender systems. 2018.
[2] Zheng, Yu, et al. "Disentangling user interest and conformity for recommendation with causal embedding." Proceedings of the Web Conference 2021. 2021.
[3] Zhang, Yang, et al. "Causal intervention for leveraging popularity bias in recommendation." Proceedings of the 44th International ACM SIGIR Conference on Research and Development in Information Retrieval. 2021.

**Questions:**

1. The authors only experiment with one backbone model, namely DLCE. I suggest the authors to compare with more state-of-the-art causal recommendation approaches.
2. What are the main contributions of the proposed model? What are the differences between the proposed pairwise relationship and existing works such as [1] and [2].
3. I suggest the authors to conduct experiments on large-scale datasets.
4. Please provide more explanations on the case study.

---

> ### Author Rebuttal · Authors · 2023-08-08
>
> Thanks for the valuable comments from this reviewer. we will give answers to each question asked by this reviewer below.
>
> **Q1:** Suggestion to compare with SOTA causal recommendation approaches.
>
> **A1:** Thanks for the suggestion. However, this review contains **factual errors** which we'd like to highlight to help reviewers better understand the goal of our paper and experiment settings.
>
> First, we clarify that our paper is **NOT** to propose a new causal recommendation approach. Instead, the proposed PropCare is a **propensity estimation** method for downstream causal recommendation, aiming to *bridge the gap in existing causality-based recommendation systems, where the propensity score and/or exposure data are often unavailable but required for model training or inference.*
>
> Second, we clarify that the term "causality-based recommendation systems" used in our paper refers to the models that recommend items with a higher causal effect (line 28-32). However, the last two papers suggested by the reviewer [b,c] are neither for propensity estimation nor for causality-based recommendation. In particular, [b] aims to tackle conformity bias by decomposing the observed ratings into factors of interest and conformity for click estimation. [c] aims to utilize desired popularity bias by adjusting them in inference for click estimation too. Only [a] is a causal recommendation model, which jointly learn two models on data with and without recommendations. Hence,  exposure data are still needed for training. The reviewer said it does not require exposure data, which is wrong. **We further report the causal recommendation results of PropCare, using CausE [a] as an alternative backbone in Table 1 of the global rebuttal PDF.** It shows a similar pattern as using DLCE backbone in the main paper, i.e., our PropCare consistently outperforms other baselines even with different causality-based models as the backbone.
>
> Third, as a popular causality-based recommendation system, DLCE [d] (RecSys'20) is still considered a SOTA causal recommendation backbone, which is also agreed by Reviewer Vbya.
>
> **Q2:**  Main contributions and comparisons with [a] and [b].
>
> **A2:** For the first contribution, please refer to the first point in our A1 to this reviewer. Our second main contribution is to *incorporate prior knowledge for robust propensity estimation* by proposing assumption 1 to model the pairwise relationship on popularity and propensity. This contribution is acknowledged by Reviewer xH3J. Moreover, the novelty of our work is also acknowledged by Reviewer 5c9m.
>
> As we have explained in the second point in A1, paper [b] is totally unrelated to our work; paper [a] is a typical causality-based recommendation system which requires exposure status to be observed for training. Our work is proposed for the gap in models like [a] where such data is usually unavailable.
>
> **Q3:** Suggestion for large-scale datasets.
>
> **A3:** Thanks for the suggestion, but other large datasets are not available for evaluation. To evaluate our model we must have ground-truth propensity and causal-effect, which are usually unavailable. Currently, only DH\_original and DH\_personalized [d] and ML [e] provide such ground-truth, which are widely used in evaluation of causality-based recommenders [d,e,f]. Nevertheless, we still conduct efficiency experiments on larger datasets, see A3 to Reviewer KaLM.
>
> **Q4:** Explanations on the case study.
>
> **A4:** The case study aims to show how the propensity scores from different approaches affect the causal recommendations. In Table 4, we investigate the top-5 recommended items with each baseline and draw the following conclusions: (1) With ground-truth propensity score and exposure, most recommended items have a positive causal effect ($\tau=1$, denoted by \ddag). (2) Comparing the lists between CJBPR and PropCare, they both hit two purchased items ($y=1$, denoted with bolded text), but CJBPR hits only one item with positive causal effect and PropCare hits two. It means although they have equal performance in conventional recommendation, PropCare outperforms CJBPR in causal recommendation. (3) Though item popularity was used to estimate propensity in some previous works [g,h,i], they are not suitable in causal recommendation.
>
> **Weakness 3:** About Tau and F1 score in Table 3.
>
> **Response**: In Table 3, we use multiple metrics to measure the accuracy of propensity estimation from different aspects. Tau only measures similarity between two rankings by counting the number of concordant and discordant pairs, which is not enough to judge whether it is a good estimation or not. Hence, we also use KLD and F1. Our PropCare performs best in KLD and F1 in most cases. Even in ML, the F1 is only 0.03 lower than the best. Moreover, using propensity and exposure inferenced by PropCare, we can achieve the best on downstream causal recommendation, which shows the advantage of PropCare over other baselines.
>
> **References**
>
> [a] Bonner, Stephen et al. "Causal embeddings for recommendation." RecSys. 2018.
>
> [b] Zheng, Yu, et al. "Disentangling user interest and conformity for recommendation with causal embedding." WWW. 2021.
>
> [c] Zhang, Yang, et al. "Causal intervention for leveraging popularity bias in recommendation." SIGIR 2021.
>
> [d] Masahiro Sato, et al. 2020. "Unbiased Learning for the Causal Effect of Recommendation". RecSys. 2020.
>
> [e] Masahiro Sato, et al. "Causality-aware neighborhood methods for recommender systems". ECIR. 2021.
>
> [f] Teng Xiao and Suhang Wang."Towards Unbiased and Robust Causal Ranking for Recommender Systems". WSDM. 2022.
>
> [g] Yuta Saito. "Unbiased Pairwise Learning from Implicit Feedback". NeurIPS 2019 Workshop on Causal Machine Learning
>
> [h] Yuta Saito, et al. "Unbiased Recommender Learning from Missing-Not-AtRandom Implicit Feedback:. WSDM. 2020.
>
> [i] Longqi Yang, et al. "Unbiased offline recommender evaluation for missing-not-at-random implicit feedback". Recsys. 2018.

---

### Official Review · Reviewer_xH3J · 2023-07-03

**Soundness:** 3 good
**Presentation:** 3 good
**Contribution:** 3 good
**Rating:** 8
**Confidence:** 4

**Summary:**

This paper proposes a framework for causality-based recommendation system. Different from traditional correlation-based recsys (e.g. collaborative filtering), causality-based recsys makes recommendations based on the causal "uplift". While there are several causal recsys models in the literature, they rely on exposure data and/or propensity scores to be given. In real-world scenarios, exposure data are often unavailable, difficult to obtain or noisy. Hence, in this paper, the authors proposed a propensity estimation framework, in the absence of exposure data which is a more practical setup, to ultimately allow causality-based recommendation. Experimental analysis are conducted with both quantitative results and a case study.

**Strengths:**

S1. The problem studied is of significant research and practical value. In particular, the setup without assuming any exposure data is practical and can be easily deployed on most recsys platforms. Overall, the motivation and challenges are well articulated and convincing.

S2. The key assumption in 4.2 is well argued and presented. The empirical validation of the assumption convincingly support the assumption.

S3. Experiments are comprehensive with evaluations on causal performance as well as the quality of estimated propensity and exposure. Detailed analysis of results reveal deeper insights, such as the factors influencing casual recommendation. Moreover, the case study is interesting and provide intuitive evidence to the benefit of a causal recsys.

S4. Overall the paper is well executed with a well motivated and effective solution.

**Weaknesses:**

W1.  Below the theoretical property in 4.5, the authors mentioned that the proposition guides certain design choices, such as regularization of the global distribution in Eq (7). While this is a theoretical implication of the proposition, is there any empirical evidence? The current ablation study does  not seem to test the usefulness of the KL regularizer.

W2. Data set differences： it is not clear to me the difference between DH_orig and DH_personalized. It was stated DH_personalized has a simulated "personalization factor". What is this factor? how does it work exactly?

Minor comments:
line 233: "should be avoid" -> avoided
Table 1: the last 4 column names, are not clearly explained.

**Questions:**

See Weaknesses.

---

> ### Author Rebuttal · Authors · 2023-08-08
>
> Thanks a lot for acknowledging the contribution of our work and paper presentation. we will give answers to each question asked by this reviewer below.
>
> **Q1:** Below the theoretical property in 4.5, the authors mentioned that the proposition guides certain design choices, such as regularization of the global distribution in Eq (7). While this is a theoretical implication of the proposition, is there any empirical evidence? The current ablation study does not seem to test the usefulness of the KL regularizer.
>
> **A1:** Sure, we have already provided additional ablation study and parameter test for KL regularizer in **Appendix D.1**, due to page limitation. We found that when $\mu =0$ (no regularization at all), the causal performance is significantly compromised. And the performance reaches the peak when $\mu$ is in the range of [0.2, 0.8], showing the advantage of regularizing the propensity with beta distribution.
>
> **Q2:** Data set differences: it is not clear to me the difference between DH\_orig and DH\_personalized. It was stated DH\_personalized has a simulated "personalization factor". What is this factor? how does it work exactly?
>
> **A2:** The difference in DH\_original and DH\_personalized is how they simulate the propensity. For DH\_original, the propensity of an item is basically computed as the ratio between how many weeks the item is exposed to the user and how many weeks the user visits the retailer. For DH\_personalized, the items are first ranked by the computed user's interaction probabilities, then the propensity is estimated by
>
> $P_{u,i}=\min \left(1, a\left(1 / \operatorname{rank}\right)^b\right),$
>
> where $\operatorname{rank}$ is the item's rank for that user $u$, $a=100$, $b=1$ are hyper-parameters. In the original paper, we refer ``personalization factor'' to the term $a(1 / \operatorname{rank})^b$. More detailed simulation steps can be found in [a].
>
> **Minor comments:** line 233: "should be avoid" - avoided Table 1: the last 4 column names, are not clearly explained.
>
> **Response to minor comments:** Thanks for pointing out these. We will revise the typo and add clear explanations for the last 4 columns of Table 1 in the latest manuscript.
>
> **reference**
>
> [a] Masahiro Sato, et al. 2020. ”Unbiased Learning for the Causal Effect of Recommendation”. RecSys. 2020.

---

### Official Review · Reviewer_Vbya · 2023-07-04

**Soundness:** 3 good
**Presentation:** 3 good
**Contribution:** 2 fair
**Rating:** 6
**Confidence:** 4

**Summary:**

The authors propose a propensity estimation/learning method based for unbiased recommendations. The method assumes no external data and only uses the user interaction data for learning. The main idea is well explained in Assumption 1 in the paper, which states that for two items with similar click/interaction probability, the more popular item is likely to be recommended to the user over the other. Authors incorporate this assumption in Eq. 6. The propensities learned via the proposed method perform better as compared to the baselines on several benchmark datasets.

**Strengths:**

- The paper is very well-written and very easy to follow. The math is also intuitive to understand, overall a good job by the authors in writing.
- The proposed method (Eq 6) follows intuitively from the main assumption (Assumption 1).
- Experiments are performed on multiple benchmark datasets, and the proposed method outperforms the baselines.
- Authors use the state-of-the-art causal recommendation method DLCE.

**Weaknesses:**

- The choice of the KL-divergence-based regularization is not super clear. The GNN paper citation (11) does not use beta distribution as a regularizer, but rather as a weight in the GNN aggregation. Also since $Q$ is the empirical distribution of the estimated propensity scores, how is the regularizer used in the training (could be explained via the gradient equation)?
- It is not clear how the ground-truth propensities in the ML dataset are used. In the appendix authors very briefly touch upon that (lines 58, 59), but it's not clear how that relates to the ground-truth propensities. MovieLens has explicit ratings, which users self-select, and the bias in the original dataset reflects the user's self-selection bias, not the recommendation bias. Saito et al [1] transform the ratings into binary feedback to model the bias used in the current paper.
-  The main metrics used for the evaluation (CP, CDCG) are not explained in the main section, but rather in the appendix. Since they are not very well-known metrics, it would be better if the authors include them in the main experimental section.

Reference:
- [1] Saito, Yuta, et al. "Unbiased recommender learning from missing-not-at-random implicit feedback." Proceedings of the 13th International Conference on Web Search and Data Mining. 2020.

**Questions:**

- What is the effect of the regularization term in propensity training? An ablation experiment could help, additionally, an experiment with varying $\mu$ values could also give some insight into this.
- If authors could elaborate more on the ML dataset setup (ground-truth propensity, binary feedback from ratings, etc)? (see point 2 in the weakness section)

**Limitations:**

Limitations are correctly addressed.

---

> ### Author Rebuttal · Authors · 2023-08-08
>
> Thanks for acknowledging the strengths of our paper. We will give answers to each question asked by this reviewer below.
>
> **Q1:** What is the effect of the regularization term in propensity training? An ablation experiment could help, additionally, an experiment with varying $\mu$ values could also give some insight into this.
>
> **A1:** The intuition of regularization term is to control the distribution of learned propensity to prevent them from clustering near extreme values like 0 and 1. In other words, to **avoid the violation of positivity assumption,** as pointed out by Reviewer 5c9m. Although the citation (cited as [11] in the main paper) does not directly use Beta distribution as a regularizer, it still uses it to control the distribution of weights. As that paper finds Beta distribution is good at modelling popularity characteristics, we chose it to model the propensity distribution which is correlated to popularity in our model.
>
> Besides, we have conducted experiments to analyze $\mu$ for the regularization term in **Appendix D.1**, due to page limitation. We found that when $\mu =0$ (no regularization at all), the causal performance is significantly compromised. And the performance reaches the peak when $\mu$ is in the range of [0.2, 0.8], showing the advantage of regularizing the propensity with beta distribution.
>
> **Q2:** If authors could elaborate more on the ML dataset setup?
>
> **A2:** Sure, let us explain it with more details. The authors of [a] also described how they pre-processed ML dataset.
>
> Given an user-item pair, first, they predicted the rating $\hat{R}$ and probabilities of observing the rating $\hat{O}$ using matrix factorization methods.
>
> Then, they estimate the interaction probability with recommendation ($\mu^{\mathrm{T}}$) and without recommendation ($\mu^{\mathrm{C}}$) using
>
> $\mu^{\mathrm{T}}=\sigma\left(\hat{R}-\epsilon\right), \quad \mu^{\mathrm{C}}=\hat{O},$
>
>  where $\sigma$ is sigmoid function and $\epsilon$ is hyper-parameter set to 5.
>
>  Next, the propensity for this user-item pair is estimated by
>
> $P=\min \left(1, a\left(1 / \operatorname{rank}\right)^b\right),$
>
> where $\operatorname{rank}$ is the rank of items for the user according to $\mu^{\mathrm{T}}+\mu^{\mathrm{C}}$. $a$ and $b$ are hyper-parameters which are set to 100 and 1 separately.
>
>  The potential outcome with and without recommendations and exposure status are sampled from their corresponding probabilities.
>
> $Y^{\mathrm{T}} \sim \operatorname{Bernoulli}\left(\mu^{\mathrm{T}}\right), \quad Y^{\mathrm{C}} \sim \operatorname{Bernoulli}\left(\mu^{\mathrm{C}}\right), \quad Z \sim \operatorname{Bernoulli}\left(P\right) .$
>
>  Finally, the causal effect and interaction are obtained as
>
> $\tau=Y^{\mathrm{T}}-Y^{\mathrm{C}}, \quad Y=Z Y^{\mathrm{T}}+\left(1-Z\right) Y^{\mathrm{C}}.$
>
> Note that in our experiment we use the obtained interaction $Y$ as ground-truth for training PropCare and causal effect $\tau$, propensity $P$, exposure status $Z$ as ground-truth for evaluation.
>
> **reference**
>
> [a] Masahiro Sato, et al. "Causality-aware neighborhood methods for recommender systems." In Advances in Information Retrieval: 43rd European Conference on IR Research, pages 603–618. Springer, 2021.

---

> > ### Comment · Reviewer_Vbya · 2023-08-17
> > **Response**
> >
> > Apologies for the late reply.
> >
> > Thanks for your clarification and response, this is helpful.

---

> > > ### Author Response · Authors · 2023-08-17
> > > **Thanks**
> > >
> > > Dear Reviewer Vbya,
> > >
> > > No worries. Thank you for letting us know that we helped.

---

### Official Review · Reviewer_KaLM · 2023-07-08

**Soundness:** 3 good
**Presentation:** 3 good
**Contribution:** 2 fair
**Rating:** 4
**Confidence:** 4

**Summary:**

This paper focuses on causality-based recommendation system, by proposing a PROPCARE method that estimates the propensity score by using its correlation with popularity. The motivation is well stated and related work is well discussed. Through experiments, the proposed method outperforms the baselines.

**Strengths:**

1. The paper is well written and easy to follow. The motivation is clearly stated and the authors did a good survey around related work.
2. The proposed method is sound, with some theoretical analysis.
3. Experiments are conducted carefully with case studies to demonstrate the effectiveness of the proposed method.

**Weaknesses:**

1. In order to model causality, the authors should make clear statement about whether the relationship between popularity and propensity is correlation or causality, which significantly affects the modeling foundation of this work.
2. Technical foundation is limited. From ML perspective, the technical contribution in this work is to combine point-wise and pair-wise modeling by combining the popularity with propensity estimation, which are all well-developed methods.
3. The baselines used in this method are not state-of-the-art, which lacks of confidence to evaluate the contribution of this work. Causality-based recommendation should share the same goal of general recommender systems, which should include the state-of-the-art methods to compare the recommendation accuracy.
4. The datasets used in this work is quite small, it's worthy to check the performance and scalability of the proposed method in large-scale datasets.

**Questions:**

1. How should the author distinguish the relationship between popularity and propensity as correlation or causality?
2. What's the recommendation accuracy comparison between  PROPCARE and state-of-the-art recommendation models?
3. How is the model's performance and scalability to large-scale datasets?

**Limitations:**

See above.

---

> ### Author Rebuttal · Authors · 2023-08-08
>
> Thanks for acknowledging our motivation, writing and experiment. we will give answers to each question asked by this reviewer below.
>
> **Q1:** Distinguish popularity and propensity as correlation or causality.
>
> **A1:** Thanks for the question. To model propensity (the probability of exposure), we first identify that item popularity directly affects the exposure status of the item, which has been also established by previous works [a,b]. Based on that fact, we propose Assumption 1 to estimate propensity using popularity as a proxy. Though the propensity is highly correlated to the popularity, it is hard to say that propensity and popularity have a direct causal relationship. First, according to interaction model stated in Eq.(2), the propensity score is also affected by interaction and relevance probabilities. Second, there might be other confounder such as item stocks that affects both propensity and popularity. Therefore, **the relationship between popularity and propensity is generally correlation, but not always be causality.** In our model (especially Assumption 1), we require the propensity and popularity to have at least correlation relationship. While two variables in a causality relationship are also correlated, it will not affect our modelling foundation whether it is causality or correlation.
>
> **Q2:** About the recommendation accuracy comparison between PROPCARE and state-of-the-art recommendation models.
> **Weakness 3:** About the baselines used in this method are not state-of-the-art. And about the same goals between causal recommendation and conventional recommendation systems.
>
> **A2 and response to weakness 3:** We have already compared the causal performance of traditional recommendation models (including **MF**, **BPR** and **LightGCN**) with PropCare (using DLCE as the causal backbone) in **Appendix D.2**, due to page limit. Here we report the comparison of traditional recommendation accuracy with LightGCN, a state-of-the-art conventional recommender below.
>
> LightGCN:
> |        | Prec@10 | Prec@100 | DCG   |
> |-----------------|---------|----------|-------|
> | DH_original     | .1029   | .0475    | 2.475 |
> | DH_personalized | .0737   | .0390    | 2.505 |
> | ML              | .3220   | .2151    | 14.78 |
>
> PropCare:
> |      | Prec@10 | Prec@100 | DCG   |
> |-----------------|---------|----------|-------|
> | DH_original     | .1967   | .0764    | 3.097 |
> | DH_personalized | .2680   | .0939    | 3.615 |
> | ML              | .6168   | .4109    | 18.21 |
>
> From the results it can be found that **our model consistently outperforms LightGCN in terms of accuracy metrics.** It is because by definition, items with high causal effect will likely be interacted when they are recommended. In contrast, items with high interaction probabilities do not necessarily have high causal effect. In other word, causality-based recommender has a similar but more focused optimization objective than conventional recommenders like LightGCN. The results also support your comment ``Causality-based recommendation should share the same goal of general recommender systems''.
>
> We clarify that as an emerging topic, there are not many approaches for estimating propensity or exposure without additional information. Previously, one popular way is to directly use item popularity (named POP in our experiment as a baseline) to simulate propensity. Besides, we also compare our method with EM and CJBPR, which are both published in recent premier conferences. There are some other works for propensity/exposure estimation, but as we have stated in the related works in the original paper, they assume observable exposure data as training labels [f,g]. For the backbone model, DLCE is also a SOTA open-sourced causal recommendation model.
>
> **Q3:** Model's performance and scalability to large-scale datasets.
>
> **A3:** It is a great suggestion to test our model on larger dataset. However, there is no other publicly available datasets for evaluation of propensity estimation or downstream causal recommendation. It is because to evaluate them we must have ground-truth propensity and causal-effect, which are usually unavailable due to privacy or technical constraints. Currently, only DH\_original and DH\_personalized provided by [c] and ML provided by [d] provide such ground-truth information. Those three datasets are widely used benchmarks in the evaluation of causality-based recommendation systems, like in [c,d,e].
>
> Nevertheless, to show the scalability of our method, we train our model on 3 popular larger-scale datasets, namely MovieLens-1M, MovieLens-10M and MovieLens-20M. We report the overhead incurred by training PropCare for propensity estimation against DLCE, the backbone causal recommender below.
>
> |    Training time (hours)      | 1M     | 10M    | 20M    |
> |----------|--------|--------|--------|
> | PropCare | .0814  | 1.494  | 3.170  |
> | DLCE     | 2.1759 | 22.658 | 40.658 |
>
>
> Note that due to the missing of ground-truth propensity/exposure and causal-effect, we are unable to evaluate the model performance. Yet, **the results show our PropCare is scalable to larger datasets and incurs only a marginal overhead on top of the backbone DLCE.**
>
> **References**
>
> [a] Zhang, Yang, et al. "Causal intervention for leveraging popularity bias in recommendation." SIGIR. 2021.
>
> [b] Tianxin Wei, et al. "Model-Agnostic Counterfactual Reasoning for Eliminating Popularity Bias in Recommender System". KDD. 2021.
>
> [c] Masahiro Sato, et al. 2020. "Unbiased Learning for the Causal Effect of Recommendation". RecSys. 2020.
>
> [d] Masahiro Sato, et al. "Causality-aware neighborhood methods for recommender systems". ECIR. 2021.
>
> [e] Teng Xiao, et al."Towards Unbiased and Robust Causal Ranking for Recommender Systems". WSDM. 2022.
>
> [f] Dawen Liang, et al. "Modeling user exposure in recommendation". WWW. 2016.
>
> [g] Masahiro Sato, et al. "Uplift-based evaluation and optimization of recommenders". RecSys, 2019

---

> > ### Comment · Reviewer_KaLM · 2023-08-18
> >
> > The reviewer would like to first thank the detailed responses from authors. Since the authors claim "While two variables in a causality relationship are also correlated, it will not affect our modelling foundation whether it is causality or correlation", the motivation and novelty of this work about causality-based recommendation is a bit misleading, the reviewer would suggest the authors to provide clear definition and empirical evidence about the causality if they would like to insist on causality-based modeling, e.g., treatment effect from purely randomized datasets, etc. Given these, the reviewer still has concerns and would like to stick with current rating.

---

> > > ### Author Response · Authors · 2023-08-19
> > > **Further response and clarification**
> > >
> > > Dear Reviewer KaLM
> > >
> > > Thanks for your response. It seems there is some misunderstandings on the direction and contribution of our work.
> > >
> > > 1.	As we have emphasized in the main paper and rebuttal to other reviewers, our main contribution is to **estimate propensity score**, which bridges the gap in existing causality-based recommendation systems, where the propensity score and/or exposure data are often unavailable but required for model training or inference. Though the downstream task is causal recommendation, please note that **we do not propose a causal recommendation model.**
> > >
> > > 2.	We would like to clarify that our work of propensity estimation **is not** building upon causality modelling. As explicitly stated in the main paper, the basis of our Assumption 1 is the intuition that, when a user’s interaction probabilities are similar toward two items i and j, but item i is more likely to be exposed to the user, the reason could be item i is more popular than j [a,b]. In accordance with this intuition, **causality between the two variables is not a requirement behind Assumption 1.** This is why we responded that "it will not affect our modelling foundation whether it is causality or correlation."
> > >
> > >
> > > In conclusion, *our research question and solution focus on the propensity estimation, instead of proposing a causal recommendation approach.*
> > >
> > > **References**
> > >
> > > [a] Zhang, Yang, et al. "Causal intervention for leveraging popularity bias in recommendation." SIGIR. 2021.
> > >
> > > [b] Tianxin Wei, et al. "Model-Agnostic Counterfactual Reasoning for Eliminating Popularity Bias in Recommender System". KDD. 2021.

---

> > > > ### Comment · Reviewer_KaLM · 2023-08-19
> > > >
> > > > Thanks for the explanation. With revisiting the paper and rebuttals, the reviewer still has concerns about the evaluation setup. Given DH datasets are still simulated but not purely randomized, it's hard to say whether it should be the source-of-truth to evaluate the performance, thus, the reviewer suggests the authors to carefully consider what's the right way to identify and model causality, rather than following existing setups from other work.

---

> > > > > ### Author Response · Authors · 2023-08-20
> > > > > **Clarification on the datasets**
> > > > >
> > > > > Dear Reviewer KaLM,
> > > > >
> > > > > Thank you for your prompt response and raising an insightful question.
> > > > >
> > > > > We agree with you that following existing methods and exiting datasets has limitations, e.g., the datasets still have some simulation elements. The reason for simulation is that exposure/propensity data are usually unknown in real-world scenarios. That is exactly why we are motivated to do this work, aiming to estimate propensity scores for downstream causal recommendation, which are usually unavailable due to privacy or technical constraints.
> > > > >
> > > > > It is a limitation that our evaluation is constrained to these datasets, but no other datasets with real-world propensity is available to the best of our knowledge. We believe this work still sheds light on an important setup toward a more practical setup for causal recommendation.
> > > > >
> > > > > We hope our comments help to answer the question.

---

### Author Rebuttal · Authors · 2023-08-08

We express our sincere gratitude to all the reviewers for their valuable feedback and insightful comments on our paper. We are humbled by the positive reception and are encouraged by the recognition of the efforts we put into this research. We acknowledge the time and expertise each reviewer has invested in evaluating our work.

Especially, we want to thank Reviewer KaLM for acknowledging our **presentation, soundness and experiments.** we thank Reviewer Vbya for  also acknowledge our **presentation, experiments and intuition.** We thank Reviewer xH3J for praising the **practical value, presentation and experiments.** We thank Reviewer Pdiw for agreeing the **proposed question, theoretical analysis and ablation study.** Finally, we thank Reviewer 5c9m for strongly appreciating the **novelty and literature review** of our work.

Please kindly find our responses to individual reviewers in the corresponding rebuttals, and some of the figures in the rebuttals can be found in the attached PDF below.

Sincerely,

The authors of "Estimating Propensity for Causality-based Recommendation without Exposure Data"

---

### Decision · Program_Chairs · 2023-09-21

**Decision:**

Accept (poster)

**Comment:**

This paper tackles the problem of propensity estimation when exposure data is not available. The main idea is by making the observation that propensity is often correlated with popularity. The estimated propensity can then be used in a downstream task, e.g., many causal-based recommender systems where propensity is required. The scores are very divided ranging from 3 to 8. Most of the reviewers agree that the problem this paper considers is very important and under-explored. The paper is clearly written and easy to understand. The reviewers also praised the paper for the theoretical analysis that is grounded with demonstratable assumptions. On the negative side, the empirical studies have to partially rely on simulated data, which hinders the credibility. I also read the paper myself and I think the reviewers are spot on with both the positive and negative sides of the paper. Overall, I still think the pros over-weigh the cons -- especially for considering a relatively underexplored (yet quite important) problem. Therefore I am voting for accept.

(For more context: the reviewer who gave the lowest score seems to misunderstand the main contribution of the paper and didn't respond to authors' rebuttal as well as the discussions among reviewers. Therefore, I didn't weigh this review as much as the others.)